# Clinical practice for migraine treatment and characteristics of medical facilities and physicians treating migraine: Insights from a retrospective cohort study using a Japanese claims database

**Tsubasa Takizawa**[1‡], **Takahiro Kitano**[2☯*], **Kanae Togo**[2☯], **Reiko Yoshikawa**[3‡], **Masahiro Iijima**[3‡]

1 Department of Neurology, Keio University School of Medicine, Tokyo, Japan, 2 Japan Access & Value, Pfizer Japan Inc., Tokyo, Japan, 3 Internal Medicine, Hospital, and Antiviral Medical Affairs, Pfizer Japan Inc., Tokyo, Japan

☯ These authors contributed equally to this work.
‡ TT, RY and MI also contributed equally to this work.
* takahiro.kitano@pfizer.com

**Data Availability Statement:** Restrictions apply to the availability of the data that supports the findings

## Abstract

The real-world treatment patterns at medical facilities and their physicians' specialties treating migraine have not been fully investigated in Japan. Therefore, a retrospective cohort study aimed to describe real-world clinical practice and treatment patterns in Japanese patients with migraine according to medical facilities and physicians' specialties. Anonymized claims data of patients with migraine was obtained from JMDC Inc (January 2018-June 2023). Patient characteristics and treatment pattern according to medical facilities and physicians' specialties treating migraine were evaluated. Of 231,156 patients with migraine (mean age [SD], 38.8 [11.8] years; females, 65.3%), 81.8% had the first prescription at clinics (CPs), 42.5% underwent imaging tests, 44.4% visited general internal medicine, and 25.9% consulted neurosurgery at initial diagnosis. Imaging tests were carried out at CPs with specialists (59.4%), hospitals (HPs) with specialists (59.1%), HPs (32.9%), and CPs (26.9%) without specialists. Overall, 95.6% received acute treatment while 21.8% received preventive treatment. At facilities with specialists compared to without specialists, triptans were more frequently prescribed (67.9% vs 44.9%) whereas acetaminophen and nonsteroidal anti-inflammatory drugs were less frequently prescribed (52.4% vs 69.2%). Preventive treatment use was higher at facilities with specialists (27.4%) than without specialists (15.7%) and increased annually regardless of the type of medical institution. In Japan, only half of patients with migraine visited facilities with specialists at their first diagnosis, and specialists are more likely use migraine-specific and preventive drugs than nonspecialists. Therefore, there is a need for awareness among migraine patients that they should consult specialists and for enhancement of medical collaboration between specialists and nonspecialists.

of this study due to contractual agreements between JMDC and hospitals. These data are available for purchase from JMDC/JMDC Claims Database (a third-party organization who owns the datasets) and qualified researchers can get access to these datasets by contacting JMDC Claims Database (website: (JP) https://www.jmdc.co.jp/jmdc-claims-database/ (ENG) https://www.jmdc.co.jp/en/jmdc-claims-database/)".

**Funding:** The funder, Pfizer Japan Inc., provided support in the form of salaries for authors [TK, KT, RY, and MI]. The medical writing support and data analytics support was also funded by Pfizer Japan Inc. (Tokyo, Japan). The funders had no additional role in study design, data collection and analysis, decision to publish, or preparation of the manuscript.

**Competing interests:** I have read the journal's policy and the authors of this manuscript have the following competing interests: TT is a consultant/advisor and/or serves on an advisory board for Pfizer, Eli Lilly, Otsuka, Amgen, and Teijin and has received speaker honoraria from Eli Lilly, Daiichi Sankyo, Otsuka, Amgen, Kowa, Kyowa Kirin, Eisai, UCB Japan, Takeda, and Santen Pharmaceutical and grant from Pfizer and research funding from Eli Lilly and Tsumura outside the submitted work. TK, KT, RY, and MI are employees of Pfizer Japan (Tokyo, Japan). KT and MI are shareholders of Pfizer Inc. The authors have no other relevant affiliations or financial involvement with any organization or entity with a financial interest in or financial conflict with the subject matter or materials discussed in the manuscript apart from those disclosed. This commercial affiliation, Pfizer Japan Inc., does not alter our adherence to PLOS ONE policies on sharing data and materials.

## Introduction

Migraine is a high-ranking contributor to the global burden of neurological disorders [1]. It is characterized by a relapsing-remitting pattern of headache of variable frequencies and more common in women than men [2, 3]. There are two major types of migraine: migraine with aura and migraine without aura. The diagnosis is based on the International Classification of Headache Disorders 3rd edition according to the patient's symptoms and characteristics [4]. Globally, migraine prevalence has been estimated to be 14–15%, and to account for 4.9% of population ill health in terms of years lived with disability [5].

The prevalence of migraine was 8.4% in Japan [6]. A substantial health and economic burden of migraine was observed in terms of decreased quality of life, impaired daily living activity, decreased work productivity/disability, and unmet needs of acute and preventive treatments of migraine in Japan [7–12].

The current acute treatments for migraine in Japan include over-the-counter (OTC; non-prescription) drugs such as combination nonsteroidal anti-inflammatory drugs (NSAIDs; acetaminophen, aspirin, and caffeine), prescription drugs such as acetaminophen, NSAIDs, triptans, antiemetics, and ergotamine [13], and lasmiditan, a selective serotonin 1F receptor agonist approved in January 2022 [14]. Preventive treatment is used for patients whose symptoms are not well managed by acute treatment to reduce the clinical, humanistic, and economic burden of the disease [15, 16]. Preventive treatment of migraine includes use of prescription drugs such as calcium channel blockers, beta-blockers, antidepressants, and anti-epileptics [17]. In addition, anti-calcitonin gene-related peptide (anti-CGRP) monoclonal antibodies (mAbs) (erenumab, galcanezumab, and fremanezumab) were approved as preventive treatments in Japan in 2021 [18–21]. While acute treatment remains the primary treatment option for migraine, the proportion of prophylactic prescriptions has been rising in recent years (2018–2022), as demonstrated in our previous retrospective cohort study on migraine treatment [22].

For the management of migraine, it is crucial to understand the role of types of medical facilities and physician specialties in migraine treatment prescriptions in Japan [10, 15]. Patients seek migraine care from primary care/internist physicians and various specialists, including neurosurgeons, general neurologists, headache specialists, and pain specialists [10, 15]. In Japan, as per the guideline for optimal use, anti-CGRP mAbs can only be prescribed at facilities that have at least one physician in charge who fulfills certain criteria including being certificated by specific academic societies [23–25]. Although the use of preventive treatment is increasing, about three out of four patients do not use preventive treatment: this may depend on the types of medical institute the patient visited in addition to the severity of migraine. Hence, it is important to understand treatment patterns of migraine in relation to characteristics of medical institutions and physicians' specialties. However, there are limited data about the characteristics of medical facilities and physicians providing migraine care, the clinical characteristics of migraine patients, and treatment patterns in relation to characteristics of medical institutions and physicians' specialties in Japan [22, 26].

In our previous study, some features related to the roles of hospitals and clinics in migraine treatment practice were described [22]. However, the relationship between physicians' specialties and the medical environment for migraine, such as the size of medical institutions and the status of imaging tests, has not been fully investigated. In addition, the pattern of migraine treatment in relation to physicians' specialties has not been examined. Insights about these issues would be useful to understand in more detail where the unmet medical needs of migraine patients lie. Hence, we conducted analysis of the retrospective claims database of Japanese patients with migraine from January 2018 to June 2023 to describe real-world clinical

practice and treatment patterns according to the types of medical facilities and physicians' specialties.

## Methods

### Study design and data source

This was a retrospective database analysis of anonymized claims data of patients from the JMDC Inc. The JMDC is a large claims database, which contains all claims data across multiple health insurance providers for company employees and their dependents [27]. It includes information sourced from inpatients, outpatients, and pharmacy claims. Within the database, individuals can be followed across multiple medical facilities and, unless they opt out of their health insurance, can be tracked even if they transfer hospitals or use multiple facilities. As of April 2024, the database had information on approximately 17 million people [28]. The database includes diagnosed disease names, coded according to Japanese Claims Codes and the International Classification of Diseases (ICD) 10th revision coding scheme, and details of prescriptions. Japan has a universal healthcare system in which the National Health Insurance covers people $\geq$ 75 years old and membership of the original health insurance society is terminated as soon as individuals reach the age of 75 years; therefore, this database does not cover claims of patients who are $\geq$ 75 years old.

### Patient selection and study period

The study period was from January 1, 2018, to June 30, 2023, and the date for data access and analysis specification finalization was November 8, 2023. Patients were included in the study if they were aged $\geq$ 18 years at the index date, had a diagnosis of migraine (ICD10: G43, excluding those with a suspicious diagnosis), and had a prescription for any migraine treatment during the study period. Patients were excluded from the study if they had < 6 months of baseline period or a diagnosis of cluster headache (ICD10: G44, excluding those with a suspicious diagnosis). The index date was the day of the first prescription for migraine treatment.

### Measurements

Study variables were as follows: patient characteristics (age at index date, sex, follow-up period, and comorbidities during baseline period); characteristics of medical facilities where the index diagnosis was made (departments, and number of beds, facilities with/without specialists according to the optimal clinical usage of guidelines for anti-CGRP mAbs) [23–25]; and status of imaging tests (computed tomography [CT] and/or magnetic resonance imaging [MRI]) performed in the 3 months before and after the initial diagnosis of migraine to exclude secondary headache/other diseases causing headache.

Medical institutions were classified as HPs (hospitals having $\geq$ 20-bed capacity), or CPs (clinics having $\leq$ 19-bed capacity) based on the Medical Care Act [29]. The medical facilities with specialists were so defined when specialists with board-certification of any of the following societies belong to the facilities: the Japanese Society of Neurology, the Japan Neurosurgical Society, the Japanese Society of Internal Medicine (specialist in general internal medicine), and the Japanese Headache Society. For the first three of these societies, the definition was based on publicly available information provided by Japan Ministry of Health, Labour and Welfare as of April 2022 [30]; for the Japanese Headache Society, the list of facilities with certified headache specialists as of May 2023 was used.

Treatment patterns included treatments prescribed (acute/preventive treatments, and treatment prescriptions from first to fourth prescriptions of migraine treatment). The drugs

prescribed for acute treatment included acetaminophen or NSAIDs, triptans, ergotamine, and lasmiditan; preventive treatment included anti-CGRP mAbs, antiepileptics, antidepressants, beta-blockers, and calcium channel blockers, which are approved for treating migraine in Japan.

## Statistical analysis

Descriptive statistics were used for the data analyses. Patient characteristics and treatment pattern were summarized using frequencies (*n*) and proportion (%). Subgroup analyses were performed according to the size of medical facilities (HP/CP), status of specialists according to the optimal clinical usage guidelines for anti-CGRP mAbs and the combination of these (HP with specialists, CP with specialists, HP without specialists, CP without specialists).

Data for treatment prescriptions were analyzed for the study period and on a yearly basis for two populations: patients with migraine included in the study, and patients who started their migraine treatment within the year of interest. Data are reported as mean and standard deviation (SD) for continuous variables, and frequency (*n*) and proportion (%) for categorical variables. Statistical analyses were performed using the SAS 9.4 (SAS Institute, Cary, NC, USA).

## Ethical consideration

This study was approved by the ethics committee of MINS (Registration No. MINS-REC-230211) as a written consent. This study was based on anonymized data: following the privacy laws and obtaining informed consent from patients was not required. The study also followed the principles outlined in the Declaration of Helsinki.

## Results

### Patient flow

Of the 15,742,853 individuals enrolled in JMDC, 2,182,587 were diagnosed with either headache or migraine during the study period. As per inclusion criteria, a total of 231,156 patients with migraine were included (Fig 1).

### Patient characteristics

The mean age (SD) of patients (*N* = 231,156) was 38.8 (11.8) years, with 65.3% females (150,861/231,156). The mean follow-up period (SD) was 26.9 (18.6) months. A total of 34.4% of patients (79,619/231,156) had underlying comorbidities during the baseline period, in which any cardiovascular-related disease (14.1%, 32,677/231,156), and neurotic, stress-related, and somatoform disorders (13.9%, 32,041/231,156) were common comorbidities (Table 1).

The majority of patients (81.8%, 189,124/231,156) received their first prescriptions of migraine treatment at CPs, and at mean age (SD) 38.8 (11.8) years, while fewer patients (18.2%, 42,102/231,156), at mean age (SD) 38.8 (12.0) years consulted HPs for their index prescriptions. Only about half of patients (48.6%, 112,332/231,156) consulted facilities with specialists while 51.4% (118,897/231,156) consulted facilities without specialists for first medical care of migraine. Similar comorbidities were observed in these subgroups by HP, CP, and facilities with or without specialists. Amongst all comorbidities, mood disorders and neurotic, stress-related, and somatoform disorders were more common among patients in the subgroup of facilities without specialists (Table 1).

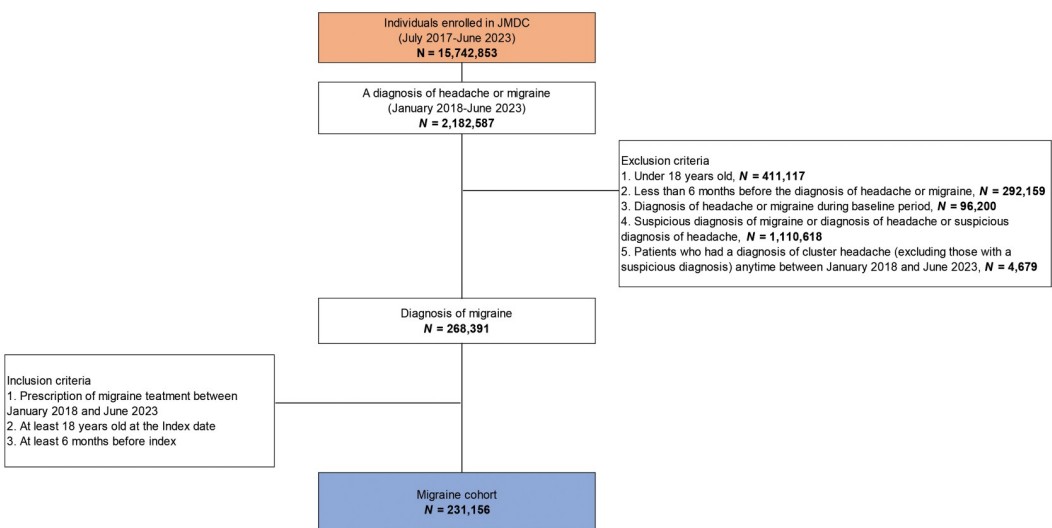

**Fig 1. Patient flow and creation of cohort.**

## Characteristics of medical facilities

Overall, 82.2% (189,958/231,156) had their first migraine diagnosis at a CP while 18.6% (42,949/231,156) received their diagnosis at an HP (Table 2).

Among the 49.1% (113,529/231,156) of patients who had their first diagnosis at facilities with specialists, 69.2% (78,561/113,529) were diagnosed at CPs. Of the 51.8% (119,653/231,156) of patients who had their first diagnosis at facilities without specialists, the majority (94.0%, 112,514/119,653) were diagnosed at CPs (Table 2).

Furthermore, 15.5% (35,809/231,156) and 3.1% (7,211/231,156) of patients had their first diagnosis at HPs with specialists and HPs without specialists, respectively, while 34.0% (78,561/231,156) and 48.7% (112,514/231,156) of patients had their first diagnosis at CPs with specialists and CPs without specialists, respectively (Table 2).

**Department of medical facilities.** Of the 231,156 patients with migraine, the majority of the total (44.4%, 102,731) and the majority who consulted HPs (25.5%, 10,931/42,942) and CPs (48.5%, 92,035/189,958) visited the general internal medicine department. Of these 231,156, patients also consulted neurosurgery (25.9%, 59,843), neurology (5.2%, 12,002), and obstetrics and gynecology (5.0%, 11,621) departments for the first diagnosis of migraine (Table 2).

Among 119,653 patients who consulted medical facilities without specialists, 51.6% (61,789) visited general internal medicine, 11.9% (14,233) visited neurosurgery, and 7.9% (9,447) visited obstetrics and gynecology departments for the first diagnosis of migraine. Further, among 113,529 patients who consulted medical facilities with specialists, 40.2% (45,680) patients visited neurosurgery, 36.3% (41,259) visited general internal medicine, and 9.2% (10,410) visited neurology departments (Table 2).

Among 78,561 patients who consulted CP with specialists, neurosurgery (47.7%, 37,496), followed by general internal medicine (40.6%, 31,911), and neurology (7.9%, 6,182) were the most visited departments. In contrast, 112,514 patients who consulted CPs without specialists, general internal medicine (53.6%, 60,313), followed by neurosurgery (12.3%, 13,835), and obstetrics and gynecology (8.2%, 9,233) were the most visited departments (Table 2).

Among 35,809 patients who consulted HPs with specialists, general internal medicine (26.4%, 9,452), neurosurgery (23.1%, 8,271), and neurology (11.8%, 4,242) were the most

**Table 1. Patient demographics and clinical characteristics.**

| Characteristics | Total | HP/CP | | Medical facilities with/without specialist | | HP/CP and with/without specialist | | | |
|---|---|---|---|---|---|---|---|---|---|
| | | HP | CP | With specialist | Without specialist | HP with specialist | HP without specialist | CP with specialist | CP without specialist |
| | $N = 231,156$ | $N = 42,102$ (18.2) | $N = 189,124$ (81.8) | $N = 112,332$ (48.6) | $N = 118,897$ (51.4) | $N = 34,964$ (15.1) | $N = 7,140$ (3.1) | $N = 77,402$ (33.5) | $N = 111,758$ (48.3) |
| Age at the index date (years) | | | | | | | | | |
| Mean | 38.8 | 38.8 | 38.8 | 38.4 | 39.2 | 38.8 | 38.5 | 38.2 | 39.2 |
| SD | 11.8 | 12.0 | 11.8 | 11.6 | 12.0 | 12.0 | 11.8 | 11.5 | 12.0 |
| Sex | | | | | | | | | |
| Male | 80,295 (34.7) | 14,017 (33.3) | 66,300 (35.1) | 39,862 (35.5) | 40,454 (34.0) | 11,998 (34.3) | 2,019 (28.3) | 27,877 (36.0) | 38,434 (34.4) |
| Female | 150,861 (65.3) | 28,085 (66.7) | 122,824 (64.9) | 72,470 (64.5) | 78,443 (66.0) | 22,966 (65.7) | 5,121 (71.7) | 49,525 (64.0) | 73,324 (65.6) |
| Follow-up period (months) | | | | | | | | | |
| Mean | 26.9 | 27.2 | 26.8 | 27.2 | 26.6 | 27.2 | 27.2 | 27.3 | 26.5 |
| SD | 18.6 | 18.5 | 18.6 | 18.6 | 18.7 | 18.6 | 18.2 | 18.6 | 18.7 |
| Comorbidities during baseline period | | | | | | | | | |
| Yes | 79,619 (34.4) | 15,120 (35.9) | 64,532 (34.1) | 37,092 (33.0) | 42,558 (35.8) | 12,541 (35.9) | 2,579 (36.1) | 24,565 (31.7) | 39,980 (35.8) |
| Mood (affective) disorders | 24,124 (10.4) | 4,257 (10.1) | 19,874 (10.5) | 9,711 (8.6) | 14,423 (12.1) | 3,221 (9.2) | 1,036 (14.5) | 6,495 (8.4) | 13,387 (12.0) |
| Neurotic, stress-related, and somatoform disorders | 32,041 (13.9) | 5,594 (13.3) | 26,466 (14.0) | 14,046 (12.5) | 18,010 (15.1) | 4,505 (12.9) | 1,089 (15.3) | 9,552 (12.3) | 16,921 (15.1) |
| Epilepsy | 4,676 (2.0) | 1,412 (3.4) | 3,265 (1.7) | 2,719 (2.4) | 1,957 (1.6) | 1,173 (3.4) | 239 (3.3) | 1,547 (2.0) | 1,718 (1.5) |
| Cerebrovascular disease | 8,909 (3.9) | 2,130 (5.1) | 6,785 (3.6) | 5,320 (4.7) | 3,592 (3.0) | 1,942 (5.6) | 188 (2.6) | 3,380 (4.4) | 3,405 (3.0) |
| Hypertension | 22,649 (9.8) | 4,551 (10.8) | 18,104 (9.6) | 10,354 (9.2) | 12,303 (10.3) | 3,816 (10.9) | 735 (10.3) | 6,540 (8.4) | 11,568 (10.4) |
| Ischemic heart diseases | 3,779 (1.6) | 929 (2.2) | 2,850 (1.5) | 1,876 (1.7) | 1,905 (1.6) | 822 (2.4) | 107 (1.5) | 1,054 (1.4) | 1,798 (1.6) |
| Peripheral vascular disease | 4,482 (1.9) | 890 (2.1) | 3,593 (1.9) | 2,080 (1.9) | 2,404 (2.0) | 761 (2.2) | 129 (1.8) | 1,319 (1.7) | 2,275 (2.0) |
| Any cardiovascular-related comorbidities listed above | 32,677 (14.1) | 6,661 (15.8) | 26,028 (13.8) | 15,847 (14.1) | 16,843 (14.2) | 5,705 (16.3) | 956 (13.4) | 10,145 (13.1) | 15,888 (14.2) |
| Malignant neoplasm of brain | 76 (0.0) | 39 (0.1) | 37 (0.0) | 57 (0.1) | 19 (0.0) | 38 (0.1) | 1 (0.0) | 19 (0.0) | 18 (0.0) |
| Malignant neoplasms (except for brain) | 4,442 (1.9) | 1,360 (3.2) | 3,085 (1.6) | 2,368 (2.1) | 2,075 (1.7) | 1,203 (3.4) | 157 (2.2) | 1,167 (1.5) | 1,918 (1.7) |
| Meningitis | 236 (0.1) | 132 (0.3) | 105 (0.1) | 162 (0.1) | 74 (0.1) | 119 (0.3) | 13 (0.2) | 44 (0.1) | 61 (0.1) |
| Disorders of thyroid gland | 10,462 (4.5) | 2,186 (5.2) | 8,279 (4.4) | 5,021 (4.5) | 5,446 (4.6) | 1,866 (5.3) | 320 (4.5) | 3,155 (4.1) | 5,127 (4.6) |
| Diabetes mellitus | 12,389 (5.4) | 2,538 (6.0) | 9,853 (5.2) | 5,570 (5.0) | 6,821 (5.7) | 2,130 (6.1) | 408 (5.7) | 3,440 (4.4) | 6,413 (5.7) |

Data are presented as *n/N* (%) unless otherwise specified.

*Abbreviations*: *CP*, clinics having $\leq$ 19-bed capacity; *HP*, hospitals having $\geq$ 20-bed capacity; *SD*, standard deviation.

**Table 2. Characteristics of medical facility and imaging tests in migraine cohort at first diagnosis of migraine.**

| Characteristics of medical facilities at the first diagnosis of migraine | Total | HP/CP | | Medical facilities with/without specialists | | HP/CP and with/without specialists | | | |
|---|---|---|---|---|---|---|---|---|---|
| | | HP | CP | With specialist | Without specialist | HP with specialist | HP without specialist | CP with specialist | CP without specialist |
| | N = 231,156 | N = 42,949 (18.6) | N = 189,958 (82.2) | N = 113,529 (49.1) | N = 119,653 (51.8) | N = 35,809 (15.5) | N = 7,211 (3.1) | N = 78,561 (34.0) | N = 112,514 (48.7) |
| Number of beds in facility | | | | | | | | | |
| 0–19 | 189,958 (82.2) | 0 (0.0) | 189,958 (100.0) | 78,561 (69.2) | 112,514 (94.0) | 0 (0.0) | 0 (0.0) | 78,561 (100.0) | 112,514 (100.0) |
| 20–99 | 9,295 (4.0) | 9,295 (21.6) | 0 (0.0) | 5,515 (4.9) | 3,783 (3.2) | 5,515 (15.4) | 3,783 (52.5) | 0 (0.0) | 0 (0.0) |
| 100–199 | 9,683 (4.2) | 9,683 (22.5) | 0 (0.0) | 7,930 (7.0) | 1,762 (1.5) | 7,930 (22.1) | 1,762 (24.4) | 0 (0.0) | 0 (0.0) |
| 200–299 | 5,017 (2.2) | 5,017 (11.7) | 0 (0.0) | 4,265 (3.8) | 752 (0.6) | 4,265 (11.9) | 752 (10.4) | 0 (0.0) | 0 (0.0) |
| 300–499 | 10,602 (4.6) | 10,602 (24.7) | 0 (0.0) | 9,939 (8.8) | 666 (0.6) | 9,939 (27.8) | 666 (9.2) | 0 (0.0) | 0 (0.0) |
| 500+ | 8,588 (3.7) | 8,588 (20.0) | 0 (0.0) | 8,339 (7.3) | 249 (0.2) | 8,339 (23.3) | 249 (3.5) | 0 (0.0) | 0 (0.0) |
| Unknown | 4 (0.0) | 0 (0.0) | 0 (0.0) | 0 (0.0) | 4 (0.0) | 0 (0.0) | 0 (0.0) | 0 (0.0) | 0 (0.0) |
| Department of medical facility[a] | | | | | | | | | |
| General internal medicine | 102,731 (44.4) | 10,931 (25.5) | 92,035 (48.5) | 41,259 (36.3) | 61,789 (51.6) | 9,452 (26.4) | 1,486 (20.6) | 31,911 (40.6) | 60,313 (53.6) |
| Neurosurgery | 59,843 (25.9) | 8,670 (20.2) | 51,297 (27.0) | 45,680 (40.2) | 14,233 (11.9) | 8,271 (23.1) | 400 (5.5) | 37,496 (47.7) | 13,835 (12.3) |
| Neurology | 12,002 (5.2) | 4,346 (10.1) | 7,679 (4.0) | 10,410 (9.2) | 1,602 (1.3) | 4,242 (11.8) | 105 (1.5) | 6,182 (7.9) | 1,497 (1.3) |
| Obstetrics and Gynecology | 11,621 (5.0) | 2,201 (5.1) | 9,423 (5.0) | 2,176 (1.9) | 9,447 (7.9) | 1,987 (5.5) | 214 (3.0) | 190 (0.2) | 9,233 (8.2) |
| Otorhinolaryngology | 5,482 (2.4) | 802 (1.9) | 4,681 (2.5) | 878 (0.8) | 4,605 (3.8) | 754 (2.1) | 48 (0.7) | 124 (0.2) | 4,557 (4.1) |
| Psychiatry | 4,938 (2.1) | 911 (2.1) | 4,228 (2.2) | 682 (0.6) | 4,457 (3.7) | 565 (1.6) | 346 (4.8) | 117 (0.1) | 4,111 (3.7) |
| Surgery | 4,181 (1.8) | 1,135 (2.6) | 3,047 (1.6) | 1,407 (1.2) | 2,775 (2.3) | 928 (2.6) | 208 (2.9) | 480 (0.6) | 2,567 (2.3) |
| Pediatrics | 4,074 (1.8) | 248 (0.6) | 3,826 (2.0) | 658 (0.6) | 3,416 (2.9) | 195 (0.5) | 53 (0.7) | 463 (0.6) | 3,363 (3.0) |
| Psychosomatic medicine | 3,473 (1.5) | 88 (0.2) | 3,385 (1.8) | 199 (0.2) | 3,274 (2.7) | 66 (0.2) | 22 (0.3) | 133 (0.2) | 3,252 (2.9) |
| Emergency | 913 (0.4) | 886 (2.1) | 27 (0.0) | 851 (0.7) | 62 (0.1) | 846 (2.4) | 40 (0.6) | 5 (0.0) | 22 (0.0) |
| Unknown | 10,078 (4.4) | 10,074 (23.5) | 0 (0.0) | 6,056 (5.3) | 4,025 (3.4) | 6,056 (16.9) | 4,021 (55.8) | 0 (0.0) | 0 (0.0) |
| Imaging tests (CT and/or MRI) at the first diagnosis of migraine | 98,127 (42.5) | 23,487 (54.7) | 76,078 (40.0) | 67,122 (59.1) | 32,558 (27.2) | 21,162 (59.1) | 2,375 (32.9) | 46,679 (59.4) | 30,234 (26.9) |

Data are presented as *n/N* (%) unless otherwise specified.

[a]Departments with ≥2% are shown.

Subgroups were defined based on the initial diagnosis of migraine for the patients included in the migraine cohort. It is possible that the same patient may visit multiple facilities in the same month, and the total of the subgroups may not match the patient number of migraine cohort.

*Abbreviations*: *CP*, clinic having ≤ 19-bed capacity; *CT*, computerized tomography; *HP*, hospital having ≥ 20-bed capacity; *MRI*, magnetic resonance imaging.

frequently visited departments. Of the 7,211 patients who consulted HPs without specialists, general internal medicine (20.6%, 1,486), neurosurgery (5.5%, 400), and psychiatry (4.8%, 346) were the most frequently visited departments (Table 2).

**Imaging tests.** Out of 231,156 patients, 42.5% (98,127) of patients underwent imaging tests (CT scans and/or MRI) before and after the initial diagnosis of migraine. The majority of patients underwent imaging tests at HPs (54.7%, 23,487) compared to CPs (40.0%, 76,078). A higher proportion of patients underwent imaging tests at CPs with specialists (59.4%, 46,679) and HPs with specialists (59.1%, 21,162) compared to patients who visited HPs without specialists (32.9%, 2,375) and CPs without specialists (26.9%, 30,234) (Table 2).

**Table 3. Treatment pattern for migraine (2018–2023).**

| Treatment | Total (2018–2023) | HP/CP | | Medical facilities with/without specialists | | HP/CP and with/without specialists | | | |
|---|---|---|---|---|---|---|---|---|---|
| | | HP | CP | With specialist | Without specialist | HP with specialist | HP without specialist | CP with specialist | CP without specialist |
| | *N* = 231,156 | *N* = 46,000 (19.9) | *N* = 192,930 (83.5) | *N* = 118,111 (51.1) | *N* = 125,385 (54.2) | *N* = 38,523 (16.7) | *N* = 7,825 (3.4) | *N* = 83,044 (35.9) | *N* = 118,191 (51.1) |
| Acute treatment | 220,922 (95.6) | 44,233 (96.2) | 183,876 (95.3) | 112,555 (95.3) | 119,807 (95.6) | 36,963 (96.0) | 7,588 (97.0) | 78,771 (94.9) | 112,802 (95.4) |
| Triptan | 128,895 (55.8) | 24,525 (53.3) | 108,858 (56.4) | 80,160 (67.9) | 56,295 (44.9) | 21,877 (56.8) | 2,818 (36.0) | 60,430 (72.8) | 53,775 (45.5) |
| Ergotamine | 6,300 (2.7) | 528 (1.1) | 5,807 (3.0) | 2,016 (1.7) | 4,365 (3.5) | 357 (0.9) | 172 (2.2) | 1,663 (2.0) | 4,198 (3.6) |
| Acetaminophen and NSAIDs | 144,967 (62.7) | 31,924 (69.4) | 115,654 (59.9) | 61,860 (52.4) | 86,707 (69.2) | 25,955 (67.4) | 6,114 (78.1) | 36,824 (44.3) | 80,849 (68.4) |
| Lasmiditan | 3,773 (1.6) | 673 (1.5) | 3,117 (1.6) | 2,022 (1.7) | 1,785 (1.4) | 627 (1.6) | 49 (0.6) | 1,399 (1.7) | 1,736 (1.5) |
| Preventive treatment | 50,353 (21.8) | 8,863 (19.3) | 42,600 (22.1) | 32,305 (27.4) | 19,721 (15.7) | 7,999 (20.8) | 904 (11.6) | 24,856 (29.9) | 18,872 (16.0) |
| Anti-CGRP mAbs | 1,597 (0.7) | 531 (1.2) | 1,085 (0.6) | 1,227 (1.0) | 391 (0.3) | 504 (1.3) | 27 (0.3) | 732 (0.9) | 364 (0.3) |
| Antiepileptics | 17,337 (7.5) | 2,894 (6.3) | 14,686 (7.6) | 10,872 (9.2) | 6,823 (5.4) | 2,505 (6.5) | 400 (5.1) | 8,500 (10.2) | 6,439 (5.4) |
| Antidepressants | 9,237 (4.0) | 1,487 (3.2) | 7,867 (4.1) | 6,541 (5.5) | 2,840 (2.3) | 1,388 (3.6) | 99 (1.3) | 5,222 (6.3) | 2,744 (2.3) |
| Beta-blockers | 4,768 (2.1) | 979 (2.1) | 3,835 (2.0) | 2,705 (2.3) | 2,133 (1.7) | 860 (2.2) | 120 (1.5) | 1,867 (2.2) | 2,016 (1.7) |
| Calcium channel blockers | 30,245 (13.1) | 5,246 (11.4) | 25,546 (13.2) | 19,613 (16.6) | 11,454 (9.1) | 4,872 (12.6) | 394 (5.0) | 14,980 (18.0) | 11,087 (9.4) |
| Acute treatment only | 180,803 (78.2) | 37,137 (80.7) | 150,330 (77.9) | 85,806 (72.6) | 105,664 (84.3) | 30,524 (79.2) | 6,921 (88.4) | 58,188 (70.1) | 99,319 (84.0) |
| Preventive treatment only | 10,234 (4.4) | 1,767 (3.8) | 9,054 (4.7) | 5,556 (4.7) | 5,578 (4.4) | 1,560 (4.0) | 237 (3.0) | 4,273 (5.1) | 5,389 (4.6) |
| Acute and preventive treatment | 40,119 (17.4) | 7,096 (15.4) | 33,546 (17.4) | 26,749 (22.6) | 14,143 (11.3) | 6,439 (16.7) | 667 (8.5) | 20,583 (24.8) | 13,483 (11.4) |

Data are presented as *n*/*N* (%) unless otherwise specified.

Subgroups were defined based on the initial diagnosis of migraine for the patients included in the migraine cohort. It is possible that the same patient may visit multiple facilities in the same month, and the total of the subgroups may not match the patient number of migraine cohort.

*Abbreviations*: *Anti-CGRP mAbs*, anti-calcitonin gene-related peptide monoclonal antibodies; *CP*, clinic having ≤ 19-bed capacity; *HP*, hospital having ≥ 20-bed capacity; *NSAIDs*, nonsteroidal anti-inflammatory drugs.

## Treatment pattern

Of the 231,156 patients, 95.6% (220,922) received acute treatment while 21.8% (50,353) received preventive treatment. Overall, 78.2% (180,803/231,156) received acute treatment alone, 21.8% (50,353/231,156) received either preventive treatment alone (4.4%, 10,234/231,156) or received both acute and preventive treatments (17.4%, 40,119/231,156) (Table 3).

Overall, 220,922 patients received acute treatment: 62.7% (114,967/231,156) patients received acetaminophen and NSAIDs; 55.8% (128,895/231,156) received triptans; 2.7% (6,300/231,156) received ergotamine; and 1.6% (3,773/231,156) received lasmiditan (Table 3). Triptans were more frequently prescribed at facilities with specialists (67.9%, 80,160/118,111) compared to without specialists (44.9%, 56,295/125,385). Among specialists, more patients were prescribed triptans at CPs with specialists (72.8%, 60,430/83,044) compared to HPs with specialists (56.8%, 21,877/38,523). In contrast, acetaminophen and other NSAIDs were less frequently prescribed at facilities with specialists (52.4%, 61,860/118,111) than without specialists (69.2%, 86,707/125,385) (Table 3).

In total, 50,353 patients received preventive treatment: 13.1% (30,245/231,156) received calcium channel blockers; 7.5% (17,337/231,156) received antiepileptic drugs; 4.0% (9,237/

231,156) received antidepressants; 2.1% (4,768/231,156) received beta-blockers; and 0.7% (1,597/231,156) received anti-CGRP mAbs (Table 3). A higher proportion of patients received preventive treatment at facilities with specialists (27.4%, 32,305/118,111) compared to without specialists (15.7%, 19,721/125,385). However, such a difference in the use of preventive treatment was not observed between CPs (22.1%, 42,600/192,930) and HPs (19.3%, 8,863/46,000). Moreover, preventive treatment use was higher at CPs with specialists (29.9%, 24,856/83,044), compared to HPs with specialists (20.8%, 7,999/38,523), and lowest at HPs without specialists (11.6%, 904/7,825) (Table 3). Treatment patterns according to treatment types and drug classes are presented in Fig 2.

**Treatment pattern year-wise.** In patients with migraine, the use of acute treatment remained high and stable from 2018 (95.7%) to 2023 (92.2%) (Fig 3 and S1 Table).

The use of preventive treatment increased in patients with migraine from 16.2% in 2018 to 28.4% in 2023. Furthermore, among preventive treatments, the use of calcium channel blockers remained high across these years (2018–2023). A similar trend was observed among all medical facility subgroups (S1 Table).

The use of anti-CGRP mAbs was observed since their approval in 2021 (0.5%) to 2023 (2.3%); usage was high in HPs (0.9% in 2021 to 4.2% in 2023) and HP facilities with specialists (1.0% in 2021 to 4.6% in 2023) (S1 Table).

Furthermore, treatment patterns among patients who started their migraine treatment within the year ($N = 231,156$) showed a similar increasing trend of preventive treatment (S2 Table).

## Treatment prescription

In patients with migraine, the use of acute treatment decreased from 94.1% (217,573/231,156) at first prescription to 29.7% (737/2,478) at fourth prescription. The use of preventive treatment increased from 16.2% (37,365/231,156) at first prescription to 72.1% (1,787/2,478) at fourth prescription. The use of preventive treatment increased from first to fourth prescription among HPs (12.9% [5,432/42,102] to 72.2% [439/608]) and among CPs (16.9% [31,947/189,124] to 72.1% [1,348/1,870]). Preventive treatment use was generally higher in facilities with specialists (20.7% [23,235/112,332] to 74.8% [1,253/1,675]) than facilities without specialists (11.9% [14,146/118,897] to 66.5% [534/803]) from first to fourth prescriptions (S3 Table).

## Discussion

This large database study provided four major findings. First, the majority of patients consulted CPs rather than HPs, and only about half of patients visited facilities with specialists for first medical care of migraine. Second, patients with migraine commonly visited general internal medicine departments, and of the patients who consulted CPs with specialists, most visited neurosurgery. In HPs with specialists, neurology was the most visited department. Third, in acute treatment, the use of triptan was high at medical facilities with specialists, whereas the use of acetaminophen and NSAIDs was high at facilities without specialists. Fourth, the use of preventive treatment was high at facilities with specialists, specifically in CPs with specialists, followed by HPs with specialists, although the use of preventive treatment increased year by year regardless of the type of medical institutions.

In this study, only half of migraine patients consulted facilities with specialists for the first diagnosis of migraine. These patients might have considered their symptoms as nonfatal or they could not find specialists [8, 31, 32]. These results suggest that there is the potential for underdiagnosis and/or undertreatment in Japan and a need to raise awareness in Japanese patients of the benefits of consulting a migraine specialist for better patient outcomes through

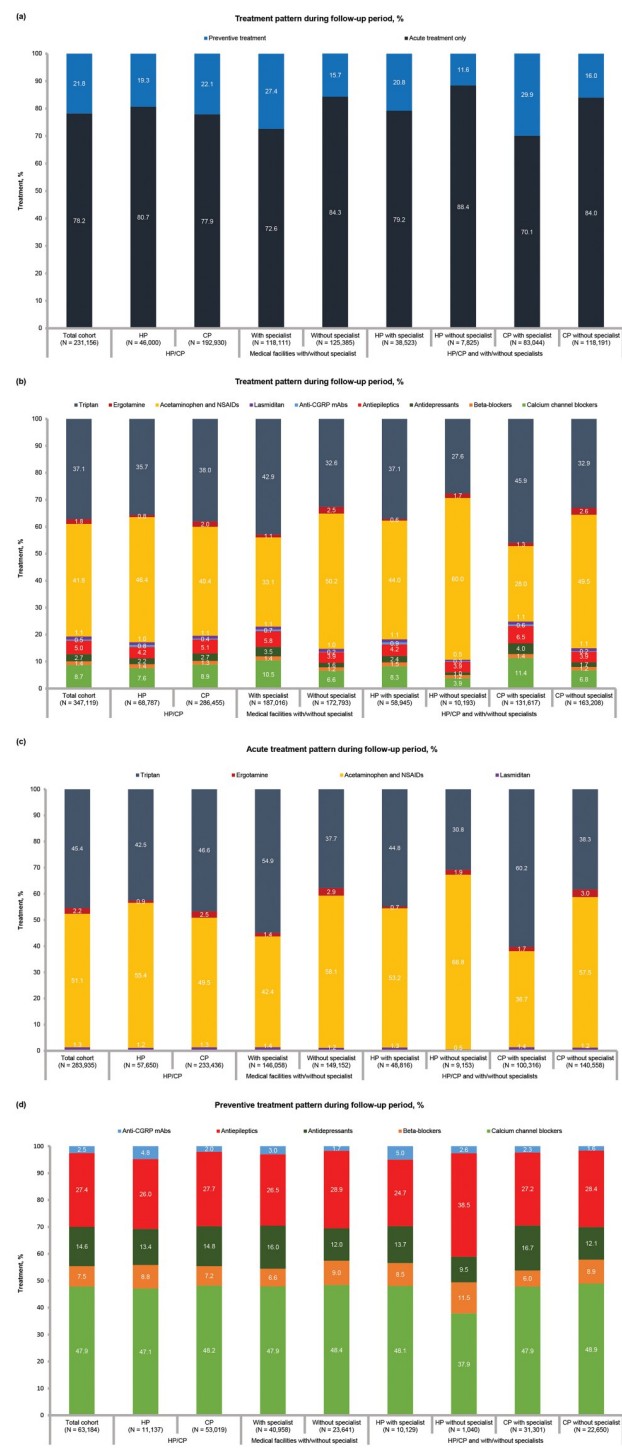

**Fig 2. Treatment pattern in total cohort and subgroups (a) type of treatments[a], (b-d) treatment drug class[b].**
[a]Preventive treatment group includes patients prescribed with acute and preventive treatments. [b]Patients prescribed with treatments from more than one drug class and counted for multiple drug classes. Subgroups were defined based on the initial diagnosis of migraine for the patients included in the migraine cohort. It is possible that the same patient may visit multiple facilities in the same month, and the total of the subgroups may not match the patient number of migraine cohort. *Abbreviations*: *Anti-CGRP mAbs*, anti-calcitonin gene-related peptide monoclonal antibodies; *CP*, clinic having ≤ 19-bed capacity; *HP*, hospital having ≥ 20-bed capacity; *NSAIDs*, nonsteroidal anti-inflammatory drugs.

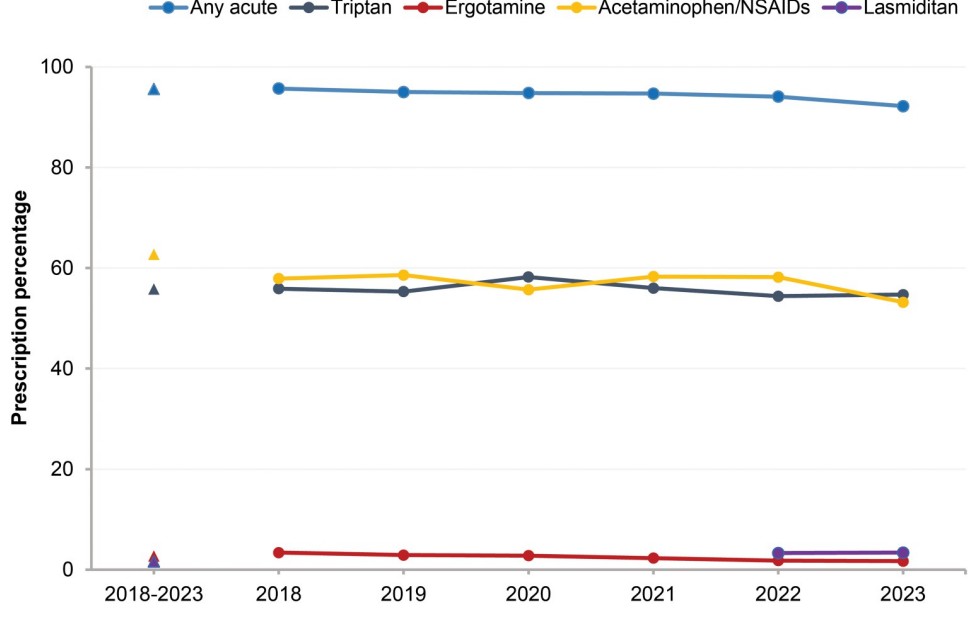

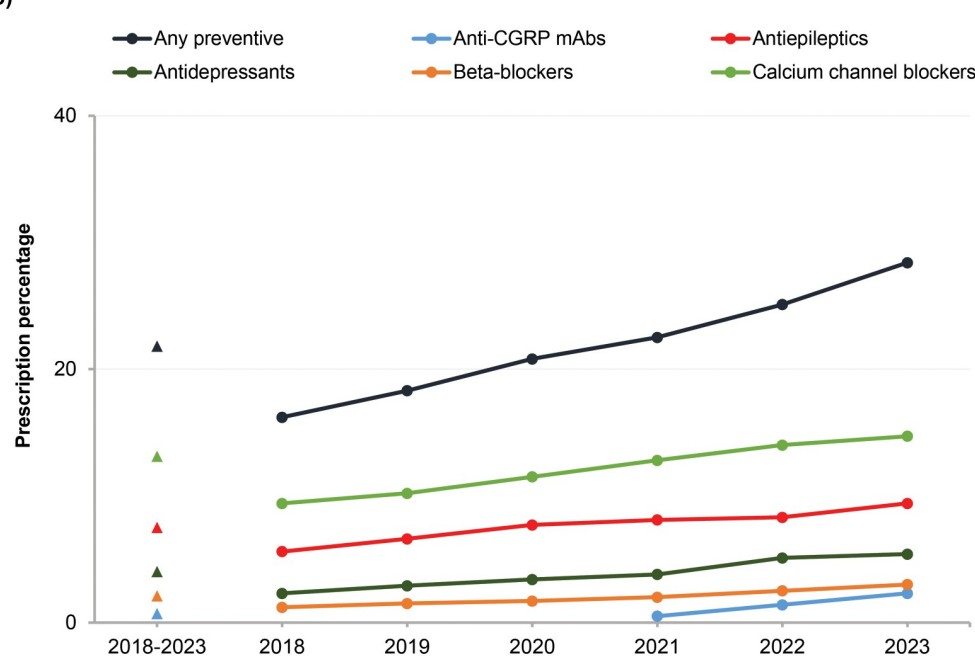

**Fig 3. Treatment pattern (2018–2023): (a) trend of acute treatment and (b) trend of preventive treatment. Number of patients: 2018–2023 (*N* = 231,156), 2018 (*n* = 39,714), 2019 (*n* = 51,116), 2020 (*n* = 56,024), 2021 (*n* = 71,582), 2022 (*n* = 74,997), 2023 (*n* = 49,751).** *Abbreviations: Anti-CGRP mAbs*, anti-calcitonin gene-related peptide monoclonal antibodies; *NSAIDs*, nonsteroidal anti-inflammatory drugs.

early effective triptan prescription in acute treatment, and appropriate use of preventive treatment [8, 15, 31]. Imaging tests were most commonly performed at CPs with specialists followed by HPs with specialists. This suggests that specialists have a thorough understanding of the standard-of-care in the clinical practice guidelines for headache disorders and are likely to actively perform imaging diagnostics for the purpose of exclusion diagnosis in patients when necessary [16].

Patients commonly visited general internal medicine departments regardless of the type of medical facility. At facilities with specialists, patients also visited neurosurgery and neurology departments at first diagnosis. In Japan, it is known that there are few neurology practitioners compared to neurosurgeons [33, 34]. There are more than 7,500 board-certified neurosurgeons and 5,000 board-certified neurologists reported in Japan [33–35], whereas the numbers of neurosurgeons vs neurologists are ~4,000 vs ~9,350 in the United States [36, 37], ~300 vs ~475 in Canada [38, 39], and ~10,715 vs ~45,000 in the European Union [40, 41], respectively. The finding of this study reflected the unique clinical practice of migraine treatment in Japan. In the United States, neurosurgeons mainly belong to university hospitals and large medical centers, and there are very few neurosurgeons who practice privately. On the other hand, in Japan, although they perform surgery for head injuries, there are some neurosurgeons who have retired from university hospitals and mainly treat general neurological diseases such as epilepsy, dementia, and headaches; it is a unique feature that they play a leading role in the treatment of migraine. Similarly, previous observational surveys in Japan reported that patients with migraine commonly consulted neurosurgeons, neurologists, and headache and pain specialists at HPs [10, 15]. Moreover, not all patients with severe migraine consulted specialists, indicating the potential for underdiagnosis and undertreatment of migraine in Japan [15].

In this study, the proportion of patients who were prescribed only acute treatment drugs was higher at facilities without specialists, regardless of whether these facilities were HPs or CPs, compared to facilities with specialists. The use of triptans was higher and that of NSAIDs lower at facilities with specialists than those without specialists. This difference between 'with specialists' and 'without specialists' was more pronounced at CPs than HPs. This might be because specialists tend to see patients with severe migraine, and are more likely to prescribe triptans, based on their understanding of the pathogenesis of migraine. In the OVERCOME (Japan) study in 9,075 individuals with migraine (July–September 2020), triptan users (~10–32%) were more likely to have consulted specialists than those using NSAIDs (~2–19%) [15].

This study also showed that the proportion of prophylactic prescriptions has been rising from 2018 to 2023. Prescription of preventive drugs was higher for specialists, especially CPs with specialists, followed by HPs with specialists. These characteristics suggest the possibility that patients with more severe migraine are seeing specialists, and that the understanding of preventive treatment is low among nonspecialists: this in turn may affect the quality of life of patients who are not prescribed preventive drugs. In addition, without access to specialists, patients may be more likely to rely on acute treatment drugs, and these patients may be at increased risk of medication-overuse headache since it has been reported that preventive medications can reduce the risk of medication-overuse headache. Therefore, there is a need to raise patients' awareness of the importance of consulting specialists, and to strengthen medical collaboration between specialists and nonspecialists. Due to the optimal usage guidelines for anti-CGRP drugs, which recommend prescription by specialists, the number of facilities where this medication can be used may be limited. As a result, there may be differences in treatment options between facilities with specialists and those without specialists [42]. The finding that CPs with specialists filled more prescriptions for preventive drugs than HPs with specialists might be explained by the following speculations. Firstly, at CPs, headache specialists can see

migraine patients regularly as the institute is most likely specializing in headache patients; however, specialists at HPs may only be available on certain days per week when the department is open. Secondly, in CPs with specialists, it is likely that specialists are seeing patients, while in HPs, even in facilities with specialists, there is a possibility that the physicians treating headaches may include nonspecialists.

The study has some limitations that are mostly intrinsic to the study design. The use of OTC drugs, frequency of migraine attacks, and actual drug usage were either not recorded or could not be ascertained from the database. In Japan, anti-CGRP mAbs should be prescribed in facilities with specialists; hence, the numbers of existing prescriptions in facilities without specialists were very small and did not affect the interpretation of the study results. We speculate that these prescription data might have been misspecified due to inaccuracy in the publicly available information about specialists, the time-lag of data for specialties, and the claims data for prescriptions. Moreover, the facilities with specialists may include nonspecialists, which could not be ascertained from the database. There may also have been a misspecification of the department that diagnosed or prescribed migraine medication, especially when multiple departments existed in a medical facility, due to the inaccuracy of the information in the claims database. This study did not present treatment patterns for migraine stratified by severity and type of migraine. Hence, further studies that incorporate the severity of migraine as reported by patients and migraine type, and that examine the association between initial prescriptions and patients' prognoses are necessary.

## Conclusion

Our study showed that only half the patients with migraine consulted specialists at their first diagnosis. Triptan and preventive treatment were more commonly used at facilities with specialists, with the highest proportion reported at CPs with specialists. This study revealed that specialists are more likely to use migraine-specific and preventive drugs than nonspecialists; therefore, there is a need of awareness among migraine patients to consult specialists, and to enhance medical collaboration between specialists and nonspecialists.

## Supporting information

**S1 Table. Migraine treatment pattern by year (2018–2023).** Subgroups were defined based on the initial diagnosis of migraine for the patients included in the migraine cohort. It is possible that the same patient may visit multiple facilities in the same month, and the total of the subgroups may not match the patient number of migraine cohort. *Abbreviations*: *Anti-CGRP mAbs*, anti-calcitonin gene-related peptide monoclonal antibodies; *CP*, clinic having ≤ 19-bed capacity; *HP*, hospital having ≥ 20-bed capacity; *NSAIDs*, nonsteroidal anti-inflammatory drugs.
(DOCX)

**S2 Table. Treatment for migraine for patients who initiated migraine treatment within the year.** Subgroups were defined based on the initial diagnosis of migraine for the patients included in the migraine cohort. It is possible that the same patient may visit multiple facilities in the same month, and the total of the subgroups may not match the patient number of migraine cohort. *Abbreviations*: *Anti-CGRP mAbs*, anti-calcitonin gene-related peptide monoclonal antibodies; *CP*, clinic having ≤ 19-bed capacity; *HP*, hospital having ≥ 20-bed capacity; *NSAID*s, nonsteroidal anti-inflammatory drugs.
(DOCX)

**S3 Table. Treatment prescriptions in migraine cohort and subgroups.** If >1 class of treatment started on the same day, each class was counted. Subgroups were defined based on the initial diagnosis of migraine for the patients included in the migraine cohort. It is possible that the same patient may visit multiple facilities in the same month, and the total of the subgroups may not match the patient number of migraine cohort. *Abbreviations*: *Anti-CGRP mAbs*, anti-calcitonin gene-related peptide monoclonal antibodies; *CP*, clinic having ≤ 19-bed capacity; *HP*, hospital having ≥ 20-bed capacity; *NSAIDs*, non-steroidal anti-inflammatory drugs. (DOCX)

## Acknowledgments

Medical writing support was provided by Niraj Vyas, PhD and Sonali Dalwadi, PhD, CMPP™ of MedPro Clinical Research. The authors sincerely thank Takumi Tajima and Yoshimitsu Takamatsu from JMDC Inc. for conducting the analysis.

## Author Contributions

**Conceptualization:** Tsubasa Takizawa, Takahiro Kitano, Kanae Togo, Reiko Yoshikawa, Masahiro Iijima.

**Data curation:** Takahiro Kitano, Kanae Togo.

**Formal analysis:** Takahiro Kitano, Kanae Togo.

**Investigation:** Tsubasa Takizawa, Takahiro Kitano, Kanae Togo, Reiko Yoshikawa, Masahiro Iijima.

**Methodology:** Tsubasa Takizawa, Takahiro Kitano, Kanae Togo, Reiko Yoshikawa, Masahiro Iijima.

**Writing – original draft:** Tsubasa Takizawa, Takahiro Kitano, Kanae Togo, Reiko Yoshikawa, Masahiro Iijima.

**Writing – review & editing:** Tsubasa Takizawa, Takahiro Kitano, Kanae Togo, Reiko Yoshikawa, Masahiro Iijima.

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
