## [Decision Letter · Decision Letter 0]

15 Nov 2024

PONE-D-24-43022Clinical practice for migraine treatment and characteristics of medical facilities and physicians treating migraine: insights from a retrospective cohort study using a Japanese claims databasePLOS ONE

Dear Dr. Kitano,

Thank you for submitting your manuscript to PLOS ONE. After careful consideration, we feel that it has merit but does not fully meet PLOS ONE’s publication criteria as it currently stands. Therefore, we invite you to submit a revised version of the manuscript that addresses the points raised during the review process.

**ACADEMIC EDITOR: **The manuscript has some minor concerns to deal with. 

We look forward to receiving your revised manuscript.

Kind regards,

Sercan Ergün

Academic Editor

PLOS ONE

Journal Requirements:

Medical writing support was provided by Niraj Vyas, PhD and Sonali Dalwadi, PhD, CMPP™ of MedPro Clinical Research, which was funded by Pfizer Japan Inc. The authors sincerely thank Takumi Tajima and Yoshimitsu Takamatsu from JMDC Inc. for conducting the analysis, which was also funded by Pfizer Japan Inc.

 The funder, Pfizer Japan Inc., provided support in the form of salaries for authors [TK, KT, RY, and MI]. The medical writing support and data analytics support was also funded by Pfizer Japan Inc. (Tokyo, Japan). The funders had no additional role in study design, data collection and analysis, decision to publish, or preparation of the manuscript. The specific roles of these authors are articulated in the ‘author contributions’ section.

Additional Editor Comments :

The manuscript has some minor concerns to deal with.

Reviewers' comments:

Reviewer's Responses to Questions

**Comments to the Author**

1. Is the manuscript technically sound, and do the data support the conclusions?

Reviewer #1: Yes

Reviewer #2: Yes

2. Has the statistical analysis been performed appropriately and rigorously? 

Reviewer #1: Yes

Reviewer #2: Yes

3. Have the authors made all data underlying the findings in their manuscript fully available?

Reviewer #1: No

Reviewer #2: Yes

4. Is the manuscript presented in an intelligible fashion and written in standard English?

Reviewer #1: Yes

Reviewer #2: Yes

5. Review Comments to the Author

Reviewer #1: Dear authors,

This retrospective cohort study aimed to describe real-world clinical practices and treatment patterns in Japanese patients with migraine based on medical facilities and physicians’ specialties. Given that migraine is a major contributor to the global burden of neurological disorders, understanding the relationship between physicians' specialties and the medical environment for migraine, as well as the patterns of migraine treatment across different specialties, would help clarify where the unmet medical needs of migraine patients lie.

Please find my comments for a better presentation of this study.

Comment 1:

Abstract: In the results section, please mention the F/M ratio and the mean age (SD) of patients.

Comment 2:

Introduction: This section is well-written and clearly outlines the objective of the study. However, it is better to mention the diagnostic strategy and different types of migraine.

Comment 3:

Results: The results section aligns with the study's aim; however, it would be beneficial to categorize patients based on migraine type, particularly in the characteristics section. How many patients experienced aura during their first presentation?

Comment 4:

Discussion: You mentioned that preventive treatments were more commonly used in facilities with specialists. It is important to discuss the significance of this finding and the potential side effects of relying solely on acute treatments. Without access to specialists, patients may resort to using more NSAIDs, which can lead to drug-overuse headaches and other side effects which you have mentioned but please discuss more about the importance of preventive strategy.

Reviewer #2: This study provides basic information on migraine diagnosis and treatment in Japan's healthcare system by analyzing a comprehensive claims database covering more than 230,000 patients. The authors provide a balanced perspective by highlighting the study's limitations, such as missing data on migraine attack frequency and medication use frequency. Despite these limitations, this well-structured study contributes valuable guidance to advance migraine care and treatment.

6. PLOS authors have the option to publish the peer review history of their article (what does this mean?). If published, this will include your full peer review and any attached files.

Reviewer #1: No

Reviewer #2: **Yes: **Murat POLAT

---

## [Author Response · Author response to Decision Letter 0]

22 Nov 2024

Response to Editor's comments:

Journal Requirements:

Response:

We have checked and revised the format of the manuscript according to PLOS ONE style requirements.

“Medical writing support was provided by Niraj Vyas, PhD and Sonali Dalwadi, PhD, CMPP™ of MedPro Clinical Research, which was funded by Pfizer Japan Inc. The authors sincerely thank Takumi Tajima and Yoshimitsu Takamatsu from JMDC Inc. for conducting the analysis, which was also funded by Pfizer Japan Inc.”

“The funder, Pfizer Japan Inc., provided support in the form of salaries for authors [TK, KT, RY, and MI]. The medical writing support and data analytics support was also funded by Pfizer Japan Inc. (Tokyo, Japan). The funders had no additional role in study design, data collection and analysis, decision to publish, or preparation of the manuscript. The specific roles of these authors are articulated in the ‘author contributions’ section.”

Response

Thank you for your suggestion. We have removed funding information from the ‘Acknowledgement section’ and revised it as below:

“Medical writing support was provided by Niraj Vyas, PhD and Sonali Dalwadi, PhD, CMPP™ of MedPro Clinical Research. The authors sincerely thank Takumi Tajima and Yoshimitsu Takamatsu from JMDC Inc. for conducting the analysis.” (Page 23 of the tracked version of the revised manuscript file)

Funding for the medical writing support and the data analytics support is already mentioned in the current funding statement: “The funder, Pfizer Japan Inc., provided support in the form of salaries for authors [TK, KT, RY, and MI]. The medical writing support and data analytics support was also funded by Pfizer Japan Inc. (Tokyo, Japan). The funders had no additional role in study design, data collection and analysis, decision to publish, or preparation of the manuscript. The specific roles of these authors are articulated in the ‘author contributions’ section.” Therefore, we confirm that no amendment is required in the current ‘Funding statement’. 

We have also included theses information in the revision cover letter. 

3. We note that you have indicated that there are restrictions to data sharing for this study. For studies involving human research participant data or other sensitive data, we encourage authors to share de-identified or anonymized data. However, when data cannot be publicly shared for ethical reasons, we allow authors to make their data sets available upon request. For information on unacceptable data access restrictions, please see http://journals.plos.org/plosone/s/data-availability#loc-unacceptable-data-access-restrictions

Response

Thank you for your suggestion. We have already added that the datasets used in this study are available to JMDC Claims Database. The access to this dataset, qualified researcher can contact JMDC Claims Database.

We have now updated the ‘Data availability’ statement as below:

“Restrictions apply to the availability of the data that supports the findings of this study due to contractual agreements between JMDC and hospitals. These data are available for purchase from JMDC/JMDC Claims Database (a third-party organization who owns the datasets) and qualified researchers can get access to these datasets by contacting JMDC Claims Database (website: (JP) https://www.jmdc.co.jp/jmdc-claims-database/ (ENG) https://www.jmdc.co.jp/en/jmdc-claims-database/).” (Page 23 of the tracked version of the revised manuscript file)

Response

We have updated the reference citation numbers throughout the manuscript and, updated the reference list. We have checked and we confirm that the revised reference list is complete and correct.

Additional Editor Comments:

The manuscript has some minor concerns to deal with.

Reviewers' comments:

Reviewer's Responses to Questions

Comments to the Author

1. Is the manuscript technically sound, and do the data support the conclusions?

Reviewer #1: Yes

Reviewer #2: Yes

Response

Thank you for your appreciating feedback. 

2. Has the statistical analysis been performed appropriately and rigorously?

Reviewer #1: Yes

Reviewer #2: Yes

Response

Thank you for your encouraging feedback.

3. Have the authors made all data underlying the findings in their manuscript fully available?

Reviewer #1: No

Reviewer #2: Yes

Response

Thank you for your comment. We have included a revised ‘Data availability statement’ in the manuscript as below. 

“Restrictions apply to the availability of the data that supports the findings of this study due to contractual agreements between JMDC and hospitals. These data are available for purchase from JMDC/JMDC Claims Database (a third-party organization who owns the datasets) and qualified researchers can get access to these datasets by contacting JMDC Claims Database (website: (JP) https://www.jmdc.co.jp/jmdc-claims-database/ (ENG) https://www.jmdc.co.jp/en/jmdc-claims-database/).” (Page 23 of the tracked version of the revised manuscript file)

4. Is the manuscript presented in an intelligible fashion and written in standard English?

Reviewer #1: Yes

Reviewer #2: Yes

Response

Thank you for your confirmation.

5. Review Comments to the Author

Reviewer #1: 

Dear authors,

This retrospective cohort study aimed to describe real-world clinical practices and treatment patterns in Japanese patients with migraine based on medical facilities and physicians’ specialties. Given that migraine is a major contributor to the global burden of neurological disorders, understanding the relationship between physicians' specialties and the medical environment for migraine, as well as the patterns of migraine treatment across different specialties, would help clarify where the unmet medical needs of migraine patients lie.

Please find my comments for a better presentation of this study.

Response

Thank you for your review comments. We have provided point by point responses to all reviewers’ comments below. 

Comment 1:

Abstract: In the results section, please mention the F/M ratio and the mean age (SD) of patients.

Response

We appreciate your comment. We have added female/male ratio and mentioned mean (SD) age of patients in the abstract as below:

“Of 231,156 patients with migraine (mean age [SD], 38.8 [11.8] years; females, 65.3%), 81.8% had the first prescription at clinics (CPs), 42.5% underwent imaging tests, 44.4% visited general internal medicine, and 25.9% consulted neurosurgery at initial diagnosis. Imaging tests were carried out at CPs with specialists (59.4%), hospitals (HPs) with specialists (59.1%), HPs (32.9%), and CPs (26.9%) without specialists.” (Page 2 of the tracked version of the revised manuscript file) 

Comment 2:

Introduction: This section is well-written and clearly outlines the objective of the study. However, it is better to mention the diagnostic strategy and different types of migraine.

Response

Thank you for your thoughtful comment. We have added the types of migraine and its diagnosis strategy in the ‘Introduction’ section with description as below: 

“Migraine is a high-ranking contributor to the global burden of neurological disorders [1]. It is characterized by a relapsing-remitting pattern of headache of variable frequencies and more common in women than men [2, 3]. There are two major types of migraine: migraine with aura and migraine without aura. The diagnosis is based on the International Classification of Headache Disorders 3rd edition according to the patient’s symptoms and characteristics [4]. Globally, migraine prevalence has been estimated to be 14–15%, and to account for 4.9% of population ill health in terms of years lived with disability [5].” (Page 3 of the tracked version of the revised manuscript file)

Comment 3:

Results: The results section aligns with the study's aim; however, it would be beneficial to categorize patients based on migraine type, particularly in the characteristics section. How many patients experienced aura during their first presentation?

Response

Thank you for your valuable suggestion. We agree that it is critically important to describe patient characteristics by the type of migraine. In this study, we also examined the type of migraine diagnosed at the index date (the first date of migraine treatment prescription). However, we found that the diagnosis of “migraine, unspecified” accounted for 96.6% of all migraine patients, and only 2.3% and 1.1% were with “migraine without aura,” and “migraine with aura,” respectively. Therefore, we decided to focus our study on the characteristics and treatment patterns of migraine patients, regardless of the type of migraine. 

We believe that investigating patients’ characteristics by the type of migraine diagnosed will be a part of future research. We have added this and revised the ‘Discussion’ section as below:

“This study did not present treatment patterns for migraine stratified by severity and type of migraine. Hence, further studies that incorporate the severity of migraine as reported by patients and migraine type, and that examine the association between initial prescriptions and patients’ prognoses are necessary.” (Page 23 of the tracked version of the revised manuscript file)

Comment 4:

Discussion: You mentioned that preventive treatments were more commonly used in facilities with specialists. It is important to discuss the significance of this finding and the potential side effects of relying solely on acute treatments. Without access to specialists, patients may resort to using more NSAIDs, which can lead to drug-overuse headaches and other side effects which you have mentioned but please discuss more about the importance of preventive strategy.

Response

Thank you for your valuable advice. Based on the suggestion, we have revised the ‘Discussion’ section as below:

“Prescription of preventive drugs was higher for specialists, especially CPs with specialists, followed by HPs with specialists. These characteristics suggest the possibility that patients with more severe migraine are seeing specialists, and that the understanding of preventive treatment is low among nonspecialists: this in turn may affect the quality of life of patients who are not prescribed preventive drugs. In addition, without access to specialists, patients may be more likely to rely on acute treatment drugs, and these patients may be at increased risk of medication-overuse headache since it has been reported that preventive medications can reduce the risk of medication-overuse headache. Therefore, there is a need to raise patients’ awareness of the importance of consulting specialists, and to strengthen medical collaboration between specialists and nonspecialists.” (Page 22 of the tracked version of the revised manuscript file)

Reviewer #2: 

This study provides basic information on migraine diagnosis and treatment in Japan's healthcare system by analyzing a comprehensive claims database covering more tha

---

## [Editor Report · Decision Letter 1]

28 Nov 2024

Clinical practice for migraine treatment and characteristics of medical facilities and physicians treating migraine: insights from a retrospective cohort study using a Japanese claims database

PONE-D-24-43022R1

Dear Dr. Kitano,

We’re pleased to inform you that your manuscript has been judged scientifically suitable for publication and will be formally accepted for publication once it meets all outstanding technical requirements.

Kind regards,

Sercan Ergün

Academic Editor

PLOS ONE
---

## [Editor Report · Acceptance letter]

2 Dec 2024

PONE-D-24-43022R1 

PLOS ONE

Dear Dr. Kitano, 

I'm pleased to inform you that your manuscript has been deemed suitable for publication in PLOS ONE. Congratulations! Your manuscript is now being handed over to our production team.

Kind regards, 

on behalf of

Dr. Sercan Ergün 

Academic Editor

PLOS ONE